# DynTopo: Dynamic Topological Scene Graph for Robotic Autonomy in Human-Centric Environments

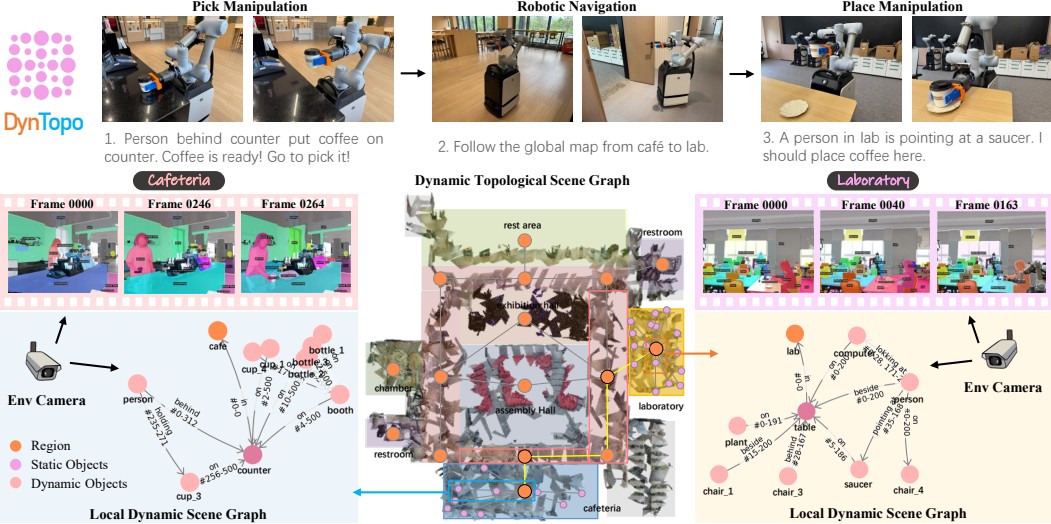

Figure 1: The proposed Dynamic Topological Scene Graph is a holistic unified graph integrating global layouts and local dynamics. Integrating such graphs into reasoning, robotics can manage autonomy in dynamic large scenes.

## Abstract

Autonomous operation of service robotics in human-centric scenes remains challenging due to the need for understanding of changing environments and context-aware decision-making. While existing approaches like topological maps offer efficient spatial priors, they fail to model transient object relationships, whereas dense neural representations (e.g., NeRF) incur prohibitive computational costs in updating. At this point, we propose the Dynamic Topological Scene Graph (DynTopo) which introduces dynamic components and relationships into persistent topological layouts for embodied robotic autonomy. Our framework constructs the global topological layouts from posed RGB-D inputs, encoding room-scale connectivity and large static objects (e.g., furniture), while environmental and egocentric cameras populate dynamic information with object position relations and human-object interaction patterns. A holistic unified architecture is conducted by integrating the dynamics into the global topology using semantic and spatial constraints, enabling seamless updates as the environment evolves. An agent powered by large language models (LLMs) is employed to interpret the unified graph, infer latent task triggers, and generate executable instructions grounded in robotic affordances. We conduct complex experiments to demonstrate DynTopo's superior scene representation effectiveness. Real-world deployments validate the system's practicality with a mobile manipulator: robotics autonomously complete complex tasks with no further training or complex rewarding in a dynamic scene as cafeteria assistant. See https://anonymous.4open.science/r/DynTopo-80C6 for video demonstration and more details.

# 1 INTRODUCTION

Recent advancements in embodied intelligence have enabled robotics to interact with complex environments Werby et al. (2024); Maggio et al. (2024); Hou et al. (2025a), yet employing them autonomously working in human-centric dynamic scenes remains challenging. A critical barrier lies in enabling robotics to (1) efficiently manage multimodal scene information, (2) reason about ongoing activities in rapidly changing environments, and (3) autonomously generate and execute tasks based on evolving environmental changes and situational awareness. Robotics struggle with the unpredictability of human-centric environments where object states, spatial relationships, and task requirements shift continuously. These limitations stem from fundamental gaps in scene-understanding architectures that fail to unify persistent scene knowledge with perceptual updates.

To conduct scene understanding, researchers have explored topological maps as memory-efficient and easily-queriable structural priors Blochliger et al. (2018); Gomez et al. (2020); Zhang (2015); Zhang et al. (2015); Garrote et al. (2018). However, such approaches exhibit critical shortcomings: Static graph nodes cannot model transient object relationships (e.g., utensils moved during cooking), while rigid hierarchies collapse under concurrent updates to entities with varying dynamism (static furniture vs. frequently manipulated items). This creates a representational mismatch between the robotics internal world model and the actual environment state, particularly in zones of high human activity. On the other hand, dense scene representations such as Neural Radiance Field (NeRF) Mildenhall et al. (2020) and Gaussian Splattings (GS) Kerbl et al. (2023) series approaches have explored introducing temporal embeddings and editing the representations to manage the dynamics Wu et al. (2024); Attal et al. (2023); Cao & Johnson (2023); Fridovich-Keil et al. (2023). However, most of these dense representations are computationally intensive to update and their dense volumetric nature hinders efficient querying for downstream tasks Hou et al. (2025a).

Recent works turn to separately represent scene components according to spatial attributes which is inspiring Hou et al. (2025a); Werby et al. (2024); Maggio et al. (2024). Topo-Field Hou et al. (2025a) leverages sparse topological map to represent the scene layouts and dense neural field for content details. Although this pipeline achieves efficiency in down-stream tasks while remains enough semantics and geometries, it can not manage dynamics based on the static scene assumption. Scene graph generation methods partially mitigate this by inferring object relationships from monocular observations Yang et al. (2023b). PSG4D Yang et al. (2023a), as an example, has demonstrated the ability to generate scene relation graph and help robotics reasoning. Yet these frameworks remain myopic because they lack mechanisms to maintain global spatial context, historical state tracking, or embodied agent perspectives. Without integrating global scene context, robotics possess limited understanding ability beyond its immediate perceptual range. Concequently, it is hard for them to execute long-horizon tasks in large-scale real-world environments.

Therefore, we aim to introduce a unified dynamic topology architecture updating as the scene evolves. Such scene representation integrates large-scale layouts with dynamic components based on spatiotemporal attributes. As shown in Fig. 1, the scene graph encodes both rarely changing elements (e.g. architectural layouts, large furniture) globally and dynamic transient objects (e.g., small items, humans), considering their evolving relationships. The dynamic topological scene graph extends the temporal expressiveness of traditional topology, while the global topology provides spatial priors to expand the perceptual horizon of localized relations. This dual enhancement equips robotics with robust autonomy in dynamically evolving scenes.

Specifically, this work proposes Dynamic Topological Scene Graph (DynTopo), a holistic unified scene graph which helps robotics understand and reason about the complex human-centric environments for self-driven embodied autonomy. From posed RGB-D inputs, we construct topological architecture capturing layout-level semantics and huge objects that are rarely moved, while localized video streams from embodied cameras or environmental cameras populate dynamic updates of the topology with object affordance and human interaction states. The topology updates are conducted according to the semantic and spatial constraints of scene components. To bring the proposed scene graph into robotic deployment, we integrate large language models (LLMs) as reasoners that interpret the unified graph, infer latent task triggers (e.g., unwashed dishes $\rightarrow$ initiate cleanup), generate executable instructions, and adapt plans as the graph evolves. The conducted DynTopo is validated through complex experiments to show its superior effectiveness. Further, real-world robotic deployments are conducted for embodied autonomy demonstration.

Our key contributions can be concluded as:

- We propose Dynamic Topological Scene Graph (DynTopo) for human-centric scene understanding. It is a holistic graph with a unified representation, which introduces dynamic scene components and relationships into topological layouts.

- We employ spatiotemporal constrained layout topology construction with open-set relation prediction and update the dynamics with geometric and semantic priors. An LLM-powered agent is employed to interpret, reason, and utilize this graph for robotic affordances.

- We conduct complex evaluations and provide real-world experimental results, demonstrating the practical viability of our approach that equips robotics with autonomy in extended dynamically evolving scenes.

## 2 RELATED WORKS

### 2.1 DYNAMIC SCENE REPRESENTATION

Representing dynamic scenes has been an essential challenging extension for scene representation. Works like T-NeRF Gao et al. (2021); Li et al. (2021; 2022); Du et al. (2021); Park et al. (2021b;a) extended NeRF Mildenhall et al. (2020) with additional time dimension or latent code. Gaussian Splatting (GS) series, as an explicit approach, also tried to adapt to dynamics Kerbl et al. (2023); Wu et al. (2024); Yang et al. (2023c); Li et al. (2024). However, dense representations are often computationally intensive and face challenges on efficient querying for downstream tasks Hou et al. (2025a). Unlike detailed scene reconstruction, topology-based representations address efficiency by abstracting environments into sparse graphs. Recent hierarchical representation approaches, such as HOV-SG Werby et al. (2024) and Topo-Field Hou et al. (2025a), introduce object-level embeddings with abstract topology to form hybrid representations. Yet based on static environments assumption, static graph vertices cannot model transient object relationships while rigid hierarchies collapse under concurrent updates to entities with varying dynamism. Our work bridges this gap through a dynamic topological scene graph that couples a persistent global topological map (encoding room layouts and static macro-objects) with dynamically updating ability based on relation prediction. This representation preserves efficiency for large-scale navigation while maintaining granular, updatable semantics for task-oriented reasoning.

### 2.2 RELATION GRAPH GENERATION

Significant progress has been made in inferring relation graphs from monocular video streams Yang et al. (2023b;a). State-of-the-art methods combine panoptic segmentation with instance tracking to detect objects and predict inter-object relationships across frames. However, these approaches remain fundamentally myopic: Their reliance on single-view inputs limits awareness of occluded regions and global spatial context, artificially constraining a robot's operational scope. For example, a robotic might recognize a "book on a desk" in its immediate view but remain oblivious to the desk's location relative to the broader home layout without scene layout knowledge. Furthermore, existing relation graph generation frameworks operate as passive observers rather than embodied agents, they lack integration with robotic action loops and have not been validated in physical task execution. To overcome these limitations, our proposed architecture introduces the dynamics (derived from egocentric or environmental camera streams) into the topological layouts. By grounding LLM-based task reasoning in this unified representation, our system not only interprets transient object relationships but also leverages persistent spatial knowledge to guide robotics through long-horizon activities.

## 3 OVERVIEW

This paper proposes Dynamic Topological scene graph to achieve embodied autonomy in dynamic environments. The scene graph can be noted as $\mathcal{G} = \{\mathcal{G}_s, \mathcal{G}_d\}$. Specifically,

$$
\begin{aligned}
\mathcal{G}_s : (\mathcal{V}_s, \mathcal{E}_s) &= \mathrm{F}(\{I_k, T_k\}_{k=1}^N), \\
\mathcal{G}_d : (\mathcal{V}_d, \mathcal{E}_d) &= \mathrm{G}(\mathcal{F}_t).
\end{aligned}
\tag{1}
$$

Figure 2: **Pipeline of our proposed Dynamic Topological Scene Graph.** The scene graph construction process consists of two branches, including the topological layouts and relation generation. A united scene graph representation is generated by integrating the semantic and geometric constraints. By employing LLM as reasoning approach, the scene graph is fed as prompts, together with other context, to drive the robotic mobile manipulator to manage task sequences.

$\mathcal{G}_s$ captures persistent environmental layouts and macro-objects which is built from posed RGB-D images $\{I_k, T_k\}_{k=1}^N (T_k \in SE(3))$, and $\mathcal{G}_d$ is the dynamic local graph incrementally built from video streams $\mathcal{F}_t$. $(\mathcal{V}, \mathcal{E})$ represents the vertices and edges in the graph. F and G denote topological layouts and dynamic relation graph construction process.

Given posed RGB-D images of the environment, the pixel-wise labels from image segmentation are back-projected to 3D space based on the corresponding depth map and camera pose to form a segmented 3D point cloud. We acquire the regions and macro-objects by querying this embedding point cloud and form the graph vertices $\mathcal{V}_s = (\mathcal{V}_r, \mathcal{V}_o)$, where $\mathcal{V}_r$ is the region vertice and $\mathcal{V}_o$ is the macro-object vertice. As for the dynamic scene graph, the process can be denoted as

$$\Pr(\mathcal{V}_d, \mathcal{E}_d \mid \mathcal{F}_t) = \Pr(M_t, O_t, R_t \mid \mathcal{F}_t), \tag{2}$$

where Pr means probability distribution, $M_t$ is the binary object mask tube, $O_t$ is the object label, and $R_t$ is the inter-object relation. The dynamic relations are anchored to the topological layouts according to semantic and spatial relations.

The scene graph serves as a structured knowledge base for LLM-based reasoning. By querying the hierarchical layers, LLM grounds with spatial and temporal context: the static graph provides global navigational constraints, while dynamic subgraphs supply localized task triggers. The LLM parses this multimodal input through prompt templates as

$$\underbrace{\text{You are} \dots}_{\text{system context}}, \underbrace{\text{Scene structures: } \mathcal{V}_s, \mathcal{E}_s}_{\text{structure from static graph}} \underbrace{\text{Ongoing relations: } \mathcal{V}_d, \mathcal{E}_d}_{\text{activities from dynamic graph}} \underbrace{\text{Optional skills: } \dots}_{\text{embodied primitives}}, \tag{3}$$

that integrate scene knowledge, skill primitives, and instructions, generating executable tasks. Eventually, they are translated into mobile base navigation and robotic arm pick-place sequences.

## 4 METHOD

Our framework operates through four core modules: topological layouts construction, dynamic relation graph generation, constrained graph fusion, and LLM-driven task reasoning. The workflow is illustrated in Fig. 2, with algorithmic details described bellow.

### 4.1 TOPOLOGICAL LAYOUTS CONSTRUCTION

The global topological layouts $\mathcal{G}_s$ capture persistent environmental semantics and represents the layout structures through vertices $\mathcal{V}_s$ and edges $\mathcal{E}_s$.

**Topology Construction.** Given posed RGB-D images $\{I_k, T_k\}_{k=1}^N (T_k \in SE(3))$, we train a Topo-Field Hou et al. (2025a) function $F : \mathbb{R}^3 \to \mathbb{R}^n$ where for any 3D point in the environment, we could access the related embeddings $\mathcal{E} \in \mathbb{R}^n$ and predict the object and region class of this specific location. We generate the object and region semantic ground-truth by applying open-vocabulary segmentation

of RGB images. The pixel-wise labels from image segmentation are back-projected to 3D space based on the corresponding depth map and camera pose to form a segmented 3D point cloud. With such trained scene representation, we could leverage it to form a topological map. Following the sample-and-query approach described in Topo-Field, we averagely sample 3D points by dividing the scene into voxel grids of 0.5m × 0.5m and regarding the center points as samples. The region and object label of points are inffered with function $F$ and form a topological map.

**Relatively Static Objects Filter.** However, current scene graph construction approaches operate under a static scene assumption, resulting in graphs that capture observational snapshots at discrete moments. Such representations inevitably suffer from transient noise and lack generalizability, including false detections and dynamic objects frequently interacting with humans. In contrast, we adopt a relatively static assumption for scene modeling, positing that scene layouts and objects exceeding specific volume thresholds (e.g., large furniture) or belonging to designated semantic categories (e.g., couch, fridge, TV) tend to remain stationary, while other entities exhibit higher dynamism. Consequently, during static scene graph construction, we selectively establish vertices only for objects with bounding-box volumes surpassing threshold $v_{thr}$ or belonging to semantic class $C_s$, effectively filtering out transient or unstable elements. After this step, a hierarchical static topological graph is built with layouts and relatively static objects:

$$\mathcal{G}_s : (\mathcal{V}_s, \mathcal{E}_s \mid \mathcal{V}_s = \{\mathbf{v}_s^r \cup \mathbf{v}_s^o\}, \mathcal{E}_s = \{\mathbf{e}_s^c \cup \mathbf{e}_s^b\}), \tag{4}$$

where $\mathcal{V}_s$ consists of region vertices $\mathbf{v}_s^r$ and static object vertices $\mathbf{v}_s^o$. $\mathcal{E}_s$ consists of region connectivity relation edges $\mathbf{e}_s^c$ and object-region belonging relation edges $\mathbf{e}_s^b$.

## 4.2 DYNAMIC RELATION GRAPH GENERATION

Dynamic relation graphs $\mathcal{G}_d$ are built and updated incrementally from video streams $\mathcal{F}_t$. The video could come from an environmental camera or an embodied one whose global pose is available.

**Video Perception and Tracking.** Given video frames set $F$, the model predicts a set of clips $\{(m_i, f_i, pr_i(c))\}_{i=1}^K$, where $m_i$ is the binary mask, $f_i$ is the related semantic feature of the specific instance, $pr_i(c)$ is the probability of assigning class $c$ to each frame in the video, and $K$ is the number of entities. We employ FC-CLIP Yu et al. (2023) as the open-set panoptic segmentation baseline to generate objects set $o_t = (m, f, c)_t$ at each frame $t$. To keep align with the on-going activities in the environment, we adapt a sliding window with a time span of $\Delta t$ to continuously track $o_t$ in each period, employing Unitrack Wang et al. (2021) as the baseline. The tracked instances during $\Delta t$ is dentoed as $\{o_j, l(o_j)\}$ where $l(o_j)$ is frame cubes when $o_j$ appears.

**Relation Prediction.** We adapt the relation prediction baseline employed in PVSG Yang et al. (2023b) to predict subject-relation-object triplets. In contrast to conventional approaches, we posit that human in the scene are more likely to act as subjects and large furniture items predominantly serve as objects. Consequently, a weight matrix is applied after the pair proposal network Wang et al. (2024a), denoted as

$$p' = \text{PPN}(f_{sub}, f_{obj}) M^{N \times N},$$

where $p'$ is the weighted pair proposals, PPN is the pair proposal network, $f_{sub}$ is the subject feature, $o_{obj}$ is the object feature, and $M$ is a weighted matrix whose element $m_{i,j}$ means the priority that object $i$ as subject and $j$ as object. In the Top-k relational pair candidates, we register wegiht of 0.7 to human as subject and large furniture as object, and weight of 0.3 to others. Then we perform relation category prediction for selected pairs $p'$. Predicted relations in $\Delta t$ period can be denoted as

$$\mathcal{R}_{\Delta t} = \{\text{id}_{sub}, \text{class}_{sub}, \text{id}_{obj}, \text{class}_{obj}, (t_a^l, t_b^l)_{l=1}^L\}_{j=1}^J,$$

where id denotes the subject / object index of instances and class is the semantic label. $(t_a^l, t_b^l)_{l=1}^L$ denotes the $L$ time spans that the relation is found. In each span, the relation happens at $t_a^l$ and ends at $t_b^l$. To enhance the temporal stability of predictions and mitigate fluctuations across time intervals, we consolidate consecutive temporal segments when the interval between the end of a preceding segment $t_b^{l-1}$ and the start of the subsequent segment $t_a^l$ is less than 2 seconds.

### 4.3 Constrained Graph Fusion

To integrate the dynamics into topological layouts, we anchor local dynamic graphs $\mathcal{G}_d$ to global topology $\mathcal{G}_s$. Depending on whether the accurate camera pose and depth is available, we provide two approaches.

**Spatial Alignments.** If the camera pose and depth is available for the environmental or embodied camera, we simply back-project the instance vertices in the dynamic graph to 3D space and merge the vertices $\mathcal{V}_s$ and $\mathcal{V}_d$ from the global and local graph whose bounding-box overlap exceeds the threshold $b_{thr}$ (60% in our experiments). For vertices in $\mathcal{V}_d$ that are not in the global graph, edges are added between them and the global region vertice, indicating the instances belong to the region.

**Semantic Matching.** If we only know the region that the camera belonging to instead of accurate camera pose and depth, we select the dynamic graph vertices $\mathcal{V}_d'$ whose predicted class belong to the designated semantic categories $C_s$, as mentioned in Section 4.1, which tends to be the relatively static stuff. We merge the connected subgraphs $\mathcal{G}_d'$ from $\mathcal{G}_d$ that $\mathcal{V}_d'$ belongs to into the global graph $\mathcal{G}_s$ and remain the related vertices and edges unchanged. For other connected subgraphs, we add an edge between the region vertice and the subgraph indicating the belonging relationship.

At each $\Delta t$ period, the dynamic components in the graph are cleared and updated as described in Section 4.2 and 4.3 to keep align with the current environment.

### 4.4 LLM-Driven Task Reasoning and Execution

To show how DynTopo can help robotics manage autonomy tasks, we introduce an LLM agent and parse the dynamic topological scene graph into textual prompts for reasoning. The LLM generates task sequences through chain-of-thought prompting and manage the tasks with navigation and object pose estimation. The pipeline is constructed as follows.

**Chain of Thoughts.** As illustrated in Equation 3, the prompts mainly consist of 1) a system instruction that describes the agent role, environment contexts, and the brief autonomy policy 2) text-formatted multilevel dynamic scene graph which is fed to the LLM at a constant frequency 3) optional skills that the robotics can manage according to the embodied ability. In each query, LLM is expected to describe the activities in the environment and conduct reasoning on whether optional skills can help with the activities. If any helpful action is available, a sequence of navigation and object pick-place tasks will be generated in order. The task instruction is formed as "navigate to / pick / place {object} in {region}" An example of prompt of a cafe assistant robotics is shown in the appendix.

**Navigation and Manipulation.** The navigation process follows a two stage planning strategy as described in ELA-ZSON Hou et al. (2025b), where the robotic approach the target instance by querying the object veretices position and planning on the graph. After approaching the target via navigation, the embodied camera takes an image and query the target object to estimate the 6 degree-of-freedom (DoF) pose. Robotic arm takes this pose as input to conduct pose-guided manipulation. The detailed manipulation process can be referenced to Polaris Wang et al. (2024b).

## 5 Experimental Results

### 5.1 Graph Structure Evaluation

Fig. 3 shows a built example of the dynamic topological scene graph in a campus building scenario, whose structure includes multiple rooms. The table shows the quantitative information of the graph vertices and edges. The evolving vertices and edges in the visualization reflect the system's capacity to adaptively update scene representations in response to environmental dynamics, demonstrating our framework's ability to maintain spatiotemporal coherence in dynamic settings.

### 5.2 Robotic Deployment

**Setups.** For environmental setup, we install environmental cameras in activity-critical zones as shown in Fig. 4. For dynamic scene graph, we activate environmental cameras to record at 5 Hz. The environmental camera pose is aligned to the unified coordinates by feeding its records to

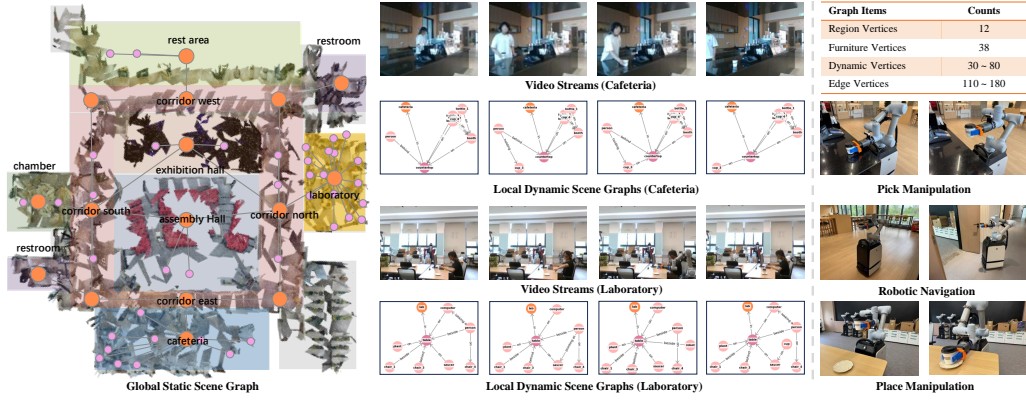

Figure 3: **Example of the generated dynamic topological scene graphs.** We show the detailed evolving process of dynamic subgraphs in the cafeteria and laboratory. On the right, we show the quantity counts of the graph vertices and edges during the process. We further show the mobile manipulation demonstrations, including navigation, pick, and place tasks.

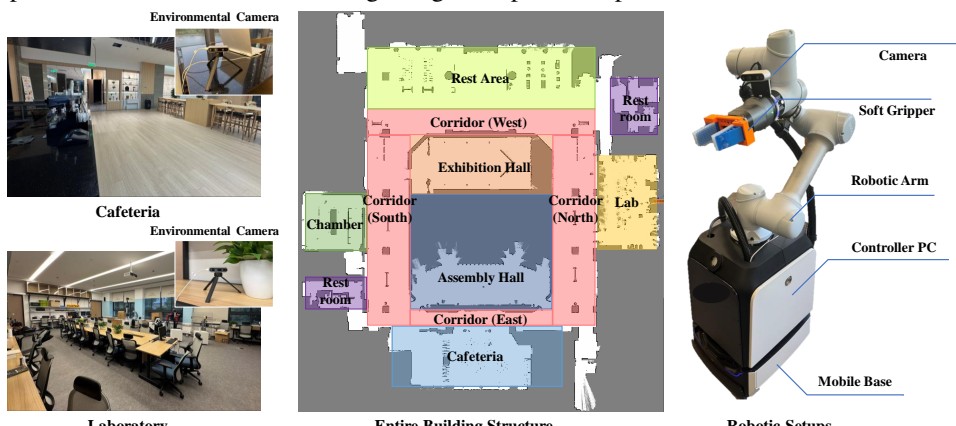

Figure 4: **Environmental and platform setups.** Environmental cameras are installed in the activity-critical regions. We show the top-down view of regions and a brief platform structure.

GLOMAP Pan et al. (2024) together with frames used to construct static graph. A 10 s sliding window is applied to extract the most recent video sequences. Our robotic platform (see Fig. 4) integrates a SLAMTEC mobile base with a SIASUN manipulator. The base employs occupancy-based mapping and navigation, while the manipulator arm hosts a RealSense D435i camera. Detailed hardware and environmental configurations are provided in the appendix.

**Demonstrations.** We provide real-world robotic deployment to show the effectiveness of the proposed method. As shown in Fig. 3, equipped with navigation and pick-place skills, the robotic acts as a cafeteria assistant in our experiments. Taking coffee delivery as an example, when someone in the laboratory places a coffee order, the environmental camera in the cafeteria monitors the beverage preparation process. Once the brewed coffee is detected on the service counter, the robot is autonomously tasked with navigating from any location to retrieve the coffee and deliver it to the laboratory. The entire workflow is performed autonomously, without manual assistance.

## 5.3 EFFECTIVENESS OF DYNAMIC COMPONENTS

**Relation Prediction.** For dynamic relation prediction, we conduct experiments on established public datasets, including Ego4D Grauman et al. (2022), Epic-Kitchens Damen et al. (2018), and VIDOR Shang et al. (2019) benchmarks originally utilized in PVSG Yang et al. (2023b). Unlike PVSG's ResNet-50 backbone, our implementation adopts the ConvNeXt-Large CLIP backbone pre-trained on LAION-2B, following the FCCLIP Yu et al. (2023) framework. We directly leverage FC-CLIP's inference strategy without fine-tuning to obtain segmentation results. For segment features tracking and relation prediction, we retain PVSG's methodology: UniTrack Wang et al.

| Methods | In-vocabulary | | | Open-vocabulary | | |
|---|---|---|---|---|---|---|
| | R/mR@20 | R/mR@50 | R/mR@100 | R/mR@20 | R/mR@50 | R/mR@100 |
| 3DSGG Wald et al. (2020) | 3.37/1.73 | 3.56/1.89 | 4.52/2.27 | 3.42/1.81 | 3.98/2.26 | 4.97/2.91 |
| PSG4D Yang et al. (2023a) | 6.15/3.46 | 6.58/4.04 | 6.83/4.51 | 6.61/3.72 | 7.02/4.48 | 7.11/4.95 |
| Ours(w/o CNN-CLIP) | 8.21/5.33 | 8.69/6.01 | 9.04/6.78 | 8.60/5.63 | 8.89/5.66 | 9.14/6.26 |
| Ours(w/o relation pair prior) | 6.12/3.41 | 7.73/5.31 | 8.02/6.54 | 9.64/6.76 | 9.82/6.95 | 9.93/7.11 |
| **DynTopo(Ours)** | **8.40/6.25** | **9.75/7.59** | **10.56/8.90** | **11.52/8.68** | **11.91/8.84** | **12.24/9.07** |

Table 1: **Quantitative comparison of relation prediction** results on OpenPVSG Yang et al. (2023b) dataset. We separately compare the in-vocabulary results and open-vocabulary results. (w/o CNN-CLIP) indicates utilizing ViT-based backbone instead of CNN-based CLIP while (w/o relation pair prior) means the relatively static scene assumption is not employed.

| Graph Approach | Methods | 0 min | | 10 min | | 20 min | | 30 min | |
|---|---|---|---|---|---|---|---|---|---|
| | | V. Acc. | E. Acc. | V. Acc. | E. Acc. | V. Acc. | E. Acc. | V. Acc. | E. Acc. |
| Static Built-from-Scratch | ConceptGraph* | 0.68 | 0.90 | 0.67 | 0.92 | 0.65 | 0.89 | 0.69 | 0.87 |
| | HOV-SG | 0.74 | **0.94** | **0.72** | **0.96** | 0.70 | **0.95** | **0.74** | **0.95** |
| | Topo-Field | **0.77** | **0.96** | 0.69 | 0.93 | **0.75** | **0.96** | 0.72 | **0.96** |
| Dynamic Updating | **DynTopo (Ours)** | **0.76** | 0.92 | **0.73** | 0.90 | **0.71** | 0.94 | 0.72 | 0.95 |

Table 2: **Quantitative comparison of multilevel dynamic graph structure** at each time interval step against static methods as time passes by (evaluated right at once, after 10 min, after 20min, and after 30 min). The V. Acc. stands for the vertices accuracy and E. Acc. stands for edges accuracy. ConceptGraph* Gu et al. (2024) indicates that, because ConceptGraph lacks explicit scene layout modeling, we substitute its layout topology with GT annotations for fair comparison. Bold and underlined numbers indicate the highest accuracy, while bold numbers represent the second highest.

(2021) associates cross-frame segmentation instances, while relation predictions derive from the candidate pair filtering and classification pipeline detailed in Section 4.2 with a transformer encoder.

We evaluate the relation prediction results and separately compare the in-vocabulary results and open-vocabulary results. The in-vocabulary results consider a prediction as accurate only if the subject category, object category, and relation is exactly same as the GT label while the open-vocabulary results consider a prediction as accurate if it predicts a reasonable result similar to the GT (e.g. person sitting on sofa v.s. adult on couch). As shown is Tab. 1, our approach performs better than previous work, especially when evaluating the open-vocabulary results.

**Evaluation on Dynamic Components.** To further validate the efficacy of our scene graph construction framework, we conduct comparative evaluations against existing static scene graph methods with the following strategy. Our multilevel dynamic scene graph is evaluated every 1 min, the graph is updated with a 10 s sliding window, while baseline methods reconstruct static scene graphs from scratch with full image sequences collected at every 1 min interval. Evaluation metrics focus on the accuracy of vertices and edges. Results in Tab. 2 indicate that our dynamic update method shows competitive performance on the updated graph structure even compared against static graph generation methods which build the graph from scratch at each time interval step.

## 5.4 Ablations

**Major Designs on Success Rate.** The real-world application success comes from the following points, and we ablate them in this section. 1) Open-set relation prediction backbone and human-centric relation priors filter. 2) Proposed components modeling strategy that separate objects into large furniture and small interactive components. 3) Dynamic subgraph updating strategy constraiend by semantic and geometry. We ablate on the success rate of the real-world experiments on the cross-room delivery task. Results are shown in Tab 3. We use origin PVSG Yang et al. (2023b) backbone when not using open-set one described in Sec. 4.2, we do not filter predicted relation pairs when not using relation pair prior in Sec. 4.2, we model all objects with static graph ignoring volume and semantic filtering when not separately modeling, we directly use dynamic subgraph to cover the related global graph nodes when not using constrained updating in Sec. 4.3.

**Object Modeling Strategy on Graph Quality.** Different from the origin static topo-graph baseline, we employ the relatively static objects filter and ablate its efficacy in helping construct robust static

| Open-set backbone | Relation pair prior | Separately modeling | Constrained updating | Coffee preparation monitoring | Coffee fetch | Delivery and place |
|---|---|---|---|---|---|---|
| ✓ | ✓ | ✓ | ✓ | 15/20 | 18/20 | 17/20 |
| ✓ | ✓ | ✓ | | 15/20 | 7/20 | 12/20 |
| ✓ | ✓ | | ✓ | - | 16/20 | - |
| ✓ | | ✓ | ✓ | 8/20 | 18/20 | 11/20 |
| | | | ✓ | 3/20 | 17/20 | 10/20 |

Table 3: **Major designs ablation on success rate of real-world object delivery task.**

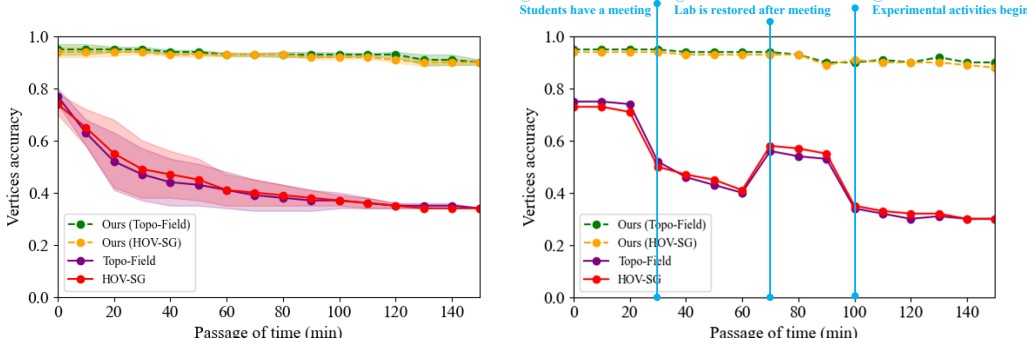

Figure 5: **Comparison of static graph vertices accuracy as time goes by.** The left plot illustrates the temporal variations in vertices accuracy across static scene graphs constructed using different methods from multiple video sequences. The right plot demonstrates the time-dependent accuracy of scene graphs in a laboratory over an extended period, with key timestamps annotated to highlight human activities that significantly impact the constructed vertices.

components. We employ HOV-SG Werby et al. (2024) and Topo-Field Hou et al. (2025a) as baselines. We recorded multiple sequences across varying times of day to capture environmental variations with various activities. The evaluated metric is the vertices precision that indicates the accuracy of the established vertice as described in ConceptGraph Gu et al. (2024). A vertice is considered correct if the predicted label is correct and overlap of predicted bounding-box and GT is more than 60%.

As time goes by, we continually compare the accuracy of the constructed vertices. The results shown in Fig. 5 indicate that our method effectively identifies and prioritizes stable scene components, specifically, the spatial distribution of functional regions and fixed macro-objects (e.g., furniture). Over extended temporal intervals, DynTopo exhibits minimal degradation from environmental dynamics, validating its robustness to transient perturbations in human-centric settings.

**Dynamic Graph Generation Strategy on Recall Rate** As shown in Tab. 1, we ablate on the segmentation backbone and the pair proposal strategy during the dynamic graph generation process. Ours(w/o CNN-CLIP) means we ablate the introduced CNN-based CLIP encoder and employ a ViT-based one. Ours(w/o relation pair prior) means we do not introduce the relatively static scene assumption that considers human as more likely to be the subject and large furniture as more likely to be the object. The results show that these strategies effectively improve performance.

## 6 CONCLUSION AND LIMITATIONS

This work introduces Dynamic Topological Scene Graph for embodied autonomy in human-centric environments. Complex experiments and real-world deployment demonstrate promising effectiveness, however, 1) While our graph architecture balances efficiency and dynamism, the sliding window for dynamic updates may introduce latency in rapidly evolving scenarios. 2) The real-world robotic deployment is realized with a junior approach by employing simple polices and given skills to ground the scene knowledge and representation. However, more powerful vision-language approaches can enable the robotic with more robust and efficient abilities.

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

## A    APPENDIX / SUPPLEMENTAL MATERIAL

### A.1    LLM USAGE AND ETHIC PROBLEM ILLUSTRATION

In the preparation of this work, the authors used GPT-4o to assist with proofreading and language polishing. This included checking for grammatical errors, improving sentence fluency, and enhancing overall clarity. After using this tool, the authors reviewed and edited the content as needed and take full responsibility for the content of the publication.

It is important to note that the core ideas, theoretical contributions, methodological design, experimental execution, data analysis, and conclusions presented in this work are solely those of the authors. The AI model was not involved in generating any of the central intellectual content.

All cameras (including embodied ones and environmental ones) underwent ethical approval, with OpenCV-based tools automatically blurring faces for privacy compliance.

### A.2    ENVIRONMENTAL SETUPS

The environment we deploy our robotics and conduct experiments is a multi-room indoor scenario of a campus building about $3029.4m^2$, consisting of a laboratory area of about $120m^2$, a cafeteria of about $440m^2$, an exhibition hall of about $164.8m^2$, an assembly hall of about $300m^2$, a rest area of about $548m^2$, a chamber of about $170m^2$, corridors of about $121.2m$ long, two restrooms, and several offices. Target manipulating objects in our experiments are chosen from the main function areas. The environmental camera is set at the cafeteria and the laboratory. The overview of the scene with exact scale is shown in Fig. A1.

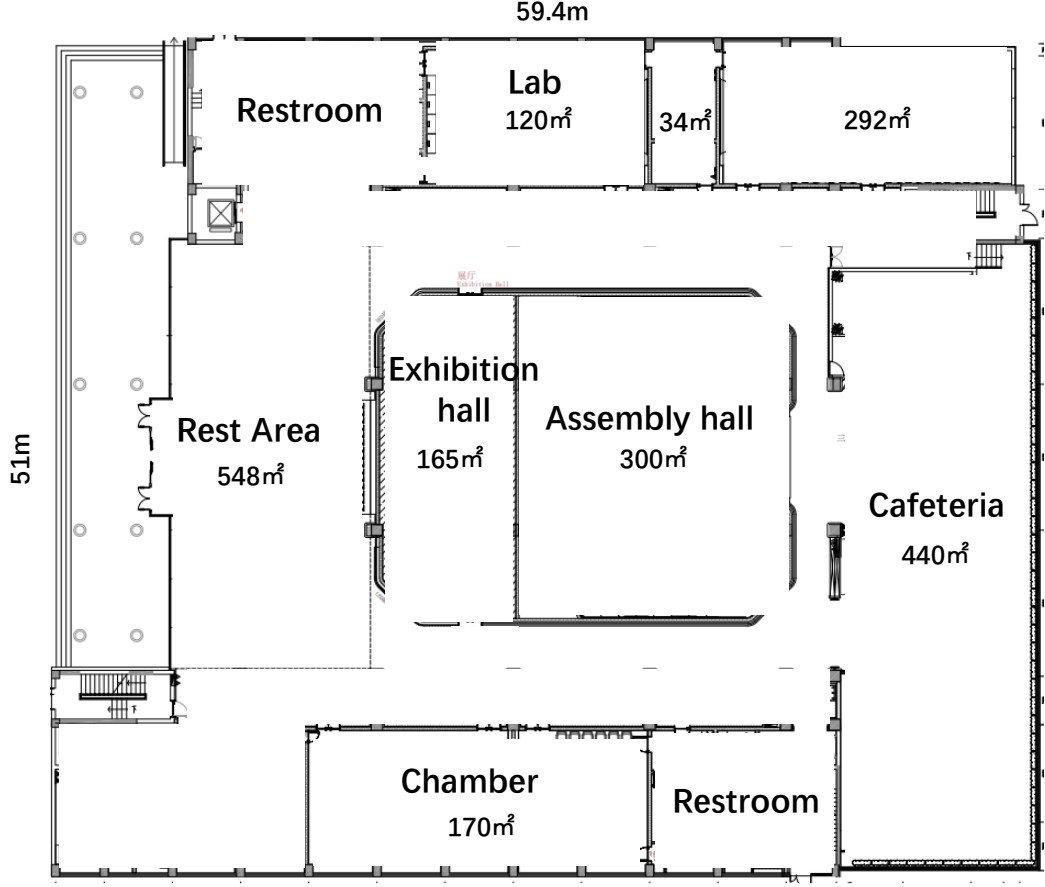

Figure A1: The top-down view of the campus building structure with the exact scale.

### A.3 ROBOTIC SETUPS

The construction of the robotic embodiment is shown in Fig.A2. We employ two types of platforms. Both mainly consist of a mobile base, a robotic arm, a Realsense D435 camera attached to the robotic arm end effector, a battery unit, and a PC. The camera is calibrated with the robotic arm base with the easy-hand-eye package. The PC is used to take control of the mobile base and get the RGB-D frame from the camera. The transformation from the arm base to the mobile base coordinates center is considered to align the RGB-D frames to the base coordinates. The maximum velocity of the mobile base is set to $1m/s$. For the mobile base, we employ the SLAMTEC Hermes, equipped with a laser radar for simplified localization and obstacle avoidance. For robotic arm, one platform utilizes the Franka panda arm, the other utilizes the SIASUN GCR5-910 arm. The graph construction, LLM reasoning, and object pose prediction algorithms are deployed on a PC equipped with and NVIDIA RTX 4090 GPU. The robotic mapping and localization, navigation, and manipulation processes are conducted on an embodied PC with an Intel i9-10885H CPU and GTX 1650ti GPU.

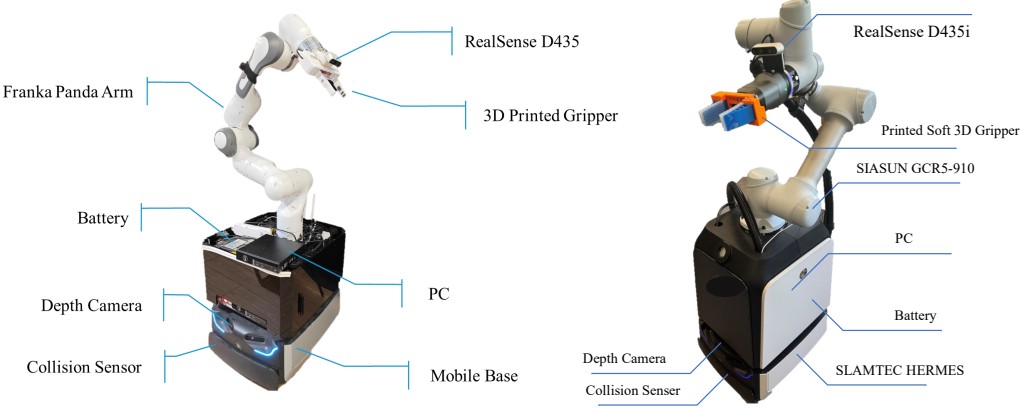

Figure A2: The hardware platform of our employed robotic mobile manipulators.

**Scene Graph Construction.** We leverage a RealSense D435i RGB-D camera to capture frames of the whole scene and employ GLOMAP Pan et al. (2024) to acquire the camera poses. The global static graph generation follows the description in Section 4.1. For dynamic scene graph, we activate environmental cameras to record at 5 Hz. The environmental camera pose is aligned to the unified coordinates by feeding its records to GLOMAP Pan et al. (2024) together with frames used to construct static graph. A 10 s sliding window is applied to extract the most recent video sequences, from which dynamic relation graphs are generated and updated as in Section 4.2.

**Robotic Manipulation.** To enable the robot to execute embodied tasks, all feasible skills are formulated as combinations of navigation and manipulator-based pick-and-place actions. For navigation, we pre-map the environment using the LiDAR on the mobile base to construct a 2D occupancy map with 5 cm resolution, ensuring localization accuracy within $\pm5$ cm. Waypoint-guided navigation is implemented via SLAMTEC's proprietary API, adhering to their documented protocols. Upon reaching target locations, the robotic arm is maneuvered to position its end-effector-mounted camera at a downward-angled viewpoint ($\approx 45°$ tilt) for optimal object observation. Captured RGB-D images feed into a 6-DoF pose estimation pipeline, with the arm executing pose-guided grasping and placement.

### A.4 MODEL SETUPS

At the static graph construction stage,

- For Topo-Field baseline, as described in their tutorial, CLIP with SwinB is employed in Detic Zhou et al. (2022), CLIP Radford et al. (2021) encoder is ViT-B/32 and Sentence-BERT Reimers & Gurevych (2019) encoder is all-mpnet-base-v2. The MHE has 18 levels of grids and the dimension of each grid is 8, with $log_2$ hash map size of 20 and only 1 hidden MLP layer of size 600.

- For HOV-SG baseline, the CLIP backbone encoder is ViT-H-14 and the SAM encoder is ViT_h_4b. The voxel size is set as $0.02m$. Other model parameters are kept align with the original setups.

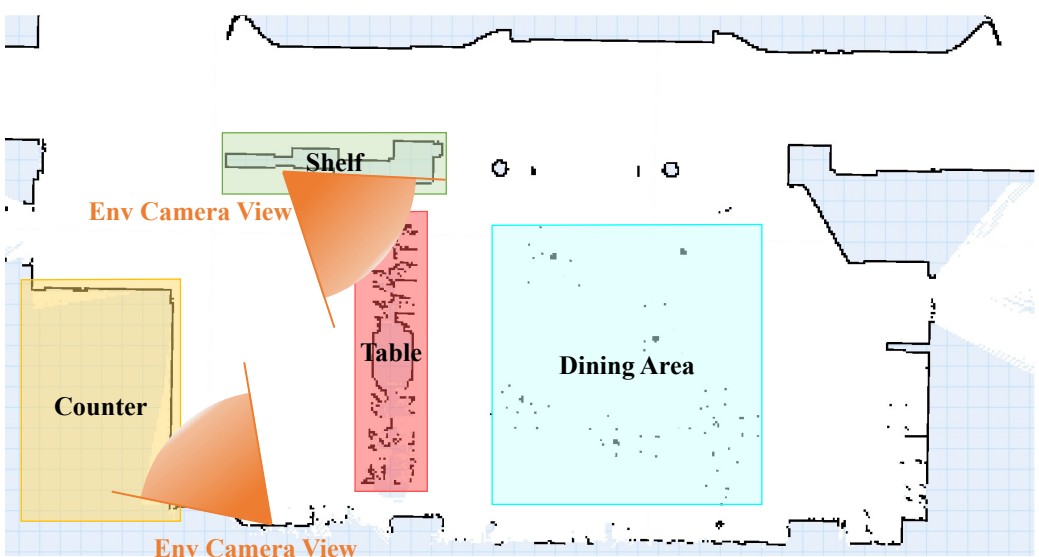

Figure A3: The top-down view of layout and environmental cameras setups of the cafeteria area.

At the dynamic graph generation stage, FC-CLIP is leveraged as segmentation baseline. The FC-CLIP employs ConvNeXt-L-d-320 as backbone which is pretrained on LAION-2B. The embedding dim is set as 768. Unitrack is employed to link segmentation results and features among frames to get the mask tubes and feature tubes. The track follows Unitrack's default settings (config/imagenet resnet18 s3 womotion.yaml) in their Github repository for Multi-Object Tracking and Segmentation (MOTS). The relation model training process follows the settings described in OpenPVSG, we employ the transformer relation head approach which performs optimal to others. As for the environmental cameras setup, we show a top-down view of the cafeteria as an example where we note the installation and view of the environmental cameras as shown in Fig. A3.

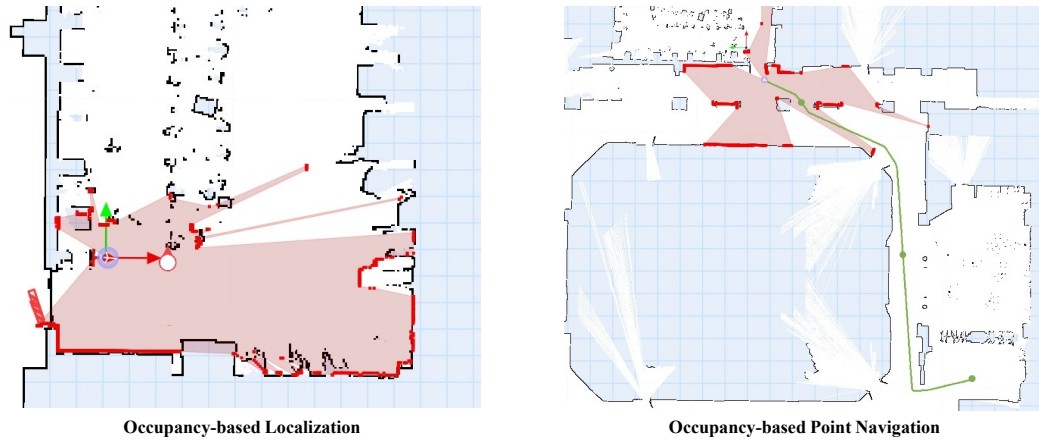

Figure A4: Examples of the occupancy-based localization and navigation of our employed mobile base.

For navigation process, we build the 2D occupancy map based on the LiDAR input with the grid size of $5cm$. Fig. A4 shows an example of the built occupancy map and occupancy matching based

localization and navigation. Consequently, the occupancy-based mapping and localization guides the pose-targeted navigation. We follow the API provided in the SLAMTEC tutorial.

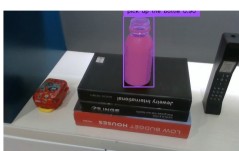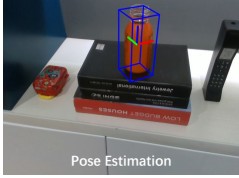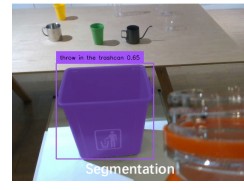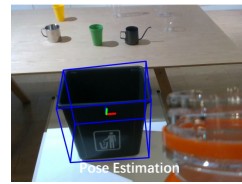

Figure A5: Medium results of our employed object pose estimation approach. It takes RGB-D image and natural language text as input and outputs the localized object with its 6-DoF pose for manipulation.

For robotic arm manipulation, we employ Polaris as the pick-place task manipulator. It takes the observation and pick-place instruction as input, leverages Grounded Light HQSAM and MVPoseNet 6D to manage the instruction grounding, and drives the robotic arm to finish the 6-DoF pose guided tasks. Fig. A5 shows the experimental medium results of the employed manipulation method, including the query and pose estimation result pairs.

## A.5 TASK VIDEOS

We provide real-world robotic embodied deployment to show the effectiveness of the proposed method. The demonstration video shows the robotics have the ability to manage the autonomy in the campus building for given high-level tasks. In the shown video, the robotic is asked to help the coffee delivery task if someone in the lab orders a coffee from the cafeteria and help keep the laboratory and cafeteria clean. Two examples are given:

- **Coffee Delivery.** As illustrated in the experiment section, The robotic is expected to help the autonomous coffee delivery process.

- **Tidy Up.** The environmental camera in the cafeteria detect the departure of patrons. The table is considered as in tidy-up needy if a person has been eating at the table and gone leaving things on the table which is not valuable. The top-down detailed map is shown in Fig. A3 of the cafeteria region.

## A.6 FAILURE CASES

Our framework, while effective in many scenarios, exhibits two primary failure modes under specific conditions.

**Relation Prediction Failures.** Relation prediction failures persist in complex or densely cluttered environments. Despite advances in scene graph generation, the relation prediction model occasionally produces inaccurate inferences or omits task-critical relationships—particularly when multiple humans interact with objects simultaneously. For instance, in crowded cafe settings, the system may fail to identify which patron has vacated their seat, leading to delayed or erroneous cleanup tasks. Similarly, if a prepared coffee is temporarily occluded (e.g., by a passing customer) when placed on the service counter, the robot may overlook the delivery trigger, leaving the item undelivered indefinitely.

**Error Recovery.** Error recovery limitations arise during long-horizon task execution. The current pipeline lacks robust mechanisms to diagnose and recover from partial failures in subtask sequences. For example, if a coffee cup grasp attempt fails due to pose estimation inaccuracies, the system does not autonomously verify subtask completion or initiate recovery protocols (e.g., re-attempting the grasp or notifying a human operator). Instead, such errors propagate silently, often resulting in full task abandonment. This limitation stems from the absence of fine-grained state verification modules and closed-loop feedback during action execution.

## A.7 LLM PROMPTS

This section provide a prompt example that is utilized to manage the robotic as a cafeteria assistant, following the approach illustrated in the methodology section as shown in 1.

Listing 1: LLM reasoning prompts example

```
{
# ===== Task Context & Role Definition =====
TASK_CONTEXT = {
    "environment": "A university academic building floor containing 1)
        Cafe area 2) Classroom cluster 3) Faculty offices 4) Common
        spaces",
    "role": "Embodied service robot for campus cafe named 'CafeBot'",
    "primary_objective": "Handle delivery tasks between cafe counter and
        various destinations while maintaining spatial awareness",
    "operational_constraints": [
        "Must navigate through mixed pedestrian-robot traffic areas",
        "Service radius limited to same-floor locations",
        "Business hours: 8:00-18:00 weekdays",
        "Priority for hot beverage deliveries under 8-minute window"
    ]
}
# ===== Multi-Level Scene Understanding =====
SCENE_UNDERSTANDING = {
    "global_topological_map": {
        "structure": "Hierarchical graph with two layer abstraction",
        "layer1_nodes": {
            "regions": ["Cafe_Station", "Classroom_A1-A6",
                "Office_B1-B4", "Elevator_Lobby", "Storage_Closet"],
            "key_objects": ["Main_Counter", "Coffee_Machine",
                "Pickup_Desk", "Emergency_Exit"]
        },
        "layer1_edges": {
            "physical_connections": [
                ("Cafe_Station <-> Elevator_Lobby via North corridor
                    (15m)"),
                ("Elevator_Lobby <-> Classroom_A1 via East hallway
                    (20m)"),
                ("Cafe_Station <-> Office_B1 via West passage (12m)")
            ],
            "semantic_links": [
                ("Coffee_Machine located_in Cafe_Station"),
                ("Pickup_Desk adjacent_to Main_Counter")
            ]
        },
        "navigation_example":
            """IF goal=Deliver to Classroom_A3:
            1. Query global topology
            2. Find Cafe_Station -> Elevator_Lobby -> Classroom_A1-A6
                cluster
            3. Calculate shortest path avoiding crowded zones during
                class breaks
            4. Update path when detecting temporary obstruction"""
    },

    "local_relational_graph": {
        "dynamic_nodes": {
            "human": ["student", "professor", "staff", "visitor"],
            "objects": {
                "static": ["table", "door", "fire_extinguisher"],
                "dynamic": ["rolling_chair", "food_tray",
                    "mobile_device"]
            }
        },
```

```
918          "relationship_edges": {
919              "spatial": ["near", "blocking", "approaching"],
920              "functional": ["waiting_for", "handing_over", "using"],
921              "temporal": ["recently_arrived", "about_to_leave"]
922          },
923          "reasoning_example":
924              """WHEN DETECTED:
925              - Barista human_node performing place_action(coffee_cup,
926                  pickup_counter)
927              - Steam rising from cup_node
928              THEN INFER:
929              1. Coffee order ready for delivery
930              2. Cup needs stabilization during transport
931              3. Priority elevation if customer waiting_time > 5min"""
932      }
933  }
934  # ===== Reasoning Chain =====
935  REASONING_MECHANISM = {
936      "core_operation_chain": [
937          "Navigate -> Grasp -> Navigate -> Place"
938      ],
939
940      "task_decomposition_examples": {
941          "coffee_delivery": {
942              "input": "Deliver coffee from counter to Classroom A3",
943              "step_breakdown": [
944                  {"action": "Navigate",
945                   "params": {"target": "Main_Counter",
946                      "path_constraints": "avoid_peak_traffic"}},
947                  {"action": "Grasp",
948                   "params": {"object": "coffee_cup", "constraints":
949                      "tilt_angle<15_degrees"}},
950                  {"action": "Navigate",
951                   "params": {"target": "Classroom_A3", "risk_avoidance":
952                      "minimize_liquid_spillage"}},
953                  {"action": "Place",
954                   "params": {"surface": "lectern_desk",
955                      "position_accuracy": "3cm"}}
956              ]
957          },
958
959          "trash_retrieval": {
960              "input": "Retrieve trash bin from Office B2",
961              "step_breakdown": [
962                  {"action": "Navigate",
963                   "params": {"target": "Office_B2", "door_operation":
964                      "auto_door_activation"}},
965                  {"action": "Grasp",
966                   "params": {"object": "trash_bin", "grip_mode":
967                      "cylindrical_grasp"}},
968                  {"action": "Navigate",
969                   "params": {"target": "Storage_Closet",
970                      "payload_awareness":
971                      "center_of_gravity_compensation"}},
972                  {"action": "Place",
973                   "params": {"surface": "recycling_zone", "orientation":
974                      "label_facing_outward"}}
975              ]
976          }
977      },
978  }
979
980  # Update PROMPT_INSTRUCTIONS
981  PROMPT_INSTRUCTIONS += f"""
```

```
4. Basic Operation Reasoning Chain:
- Mandatory decomposition structure:
    {REASONING_MECHANISM['core_operation_chain']}
- Standard workflow example:
    {REASONING_MECHANISM['task_decomposition_examples']
['coffee_delivery']['step_breakdown']}
- Error recovery protocols:
    {REASONING_MECHANISM['error_recovery_protocols']
['grasp_failure'][0:2]}

Strictly prohibited:
- Introducing action types beyond grasp/place
- Adding object modification or complex interaction
- Executing non-navigation movement commands
"""
# New minimal example
"document_transfer": {
    "input": "Transfer documents from printer room to professor's
        office",
    "step_breakdown": [
        {"action": "Navigate",
         "params": {"target": "Printer_Room", "elevator_usage":
            "priority_freight_elevator"}},
        {"action": "Grasp",
         "params": {"object": "document_stack", "pressure_control":
            "prevent_paper_crease"}},
        {"action": "Navigate",
         "params": {"target": "Professor_Office", "social_navigation":
            "avoid_private_conversation_areas"}},
        {"action": "Place",
         "params": {"surface": "incoming_tray", "alignment":
            "parallel_to_desk_edge"}}
    ]
}
# ===== Execution Protocol =====
PROMPT_INSTRUCTIONS = f"""
You are {TASK_CONTEXT['role']} operating in
    {TASK_CONTEXT['environment']}. Your core capabilities include:

1. Topological Navigation:
- Maintain mental map:
    {SCENE_UNDERSTANDING['global_topological_map']['structure']}
- Use regional connections like
    {SCENE_UNDERSTANDING['global_topological_map']
['layer1_edges']['physical_connections'][0]} for long-range planning
- Example: {SCENE_UNDERSTANDING['global_topological_map']
['navigation_example']}

2. Situational Reasoning:
- Track relationships: {SCENE_UNDERSTANDING['local_relational_graph']
['relationship_edges']['spatial']}
- Make inferences like {SCENE_UNDERSTANDING['local_relational_graph']
['reasoning_example']}
- Detect human activities (e.g., professor_chen approaching door -> hold
    door open)

3. Skill Orchestration:
- Compose primitives: {ROBOT_CAPABILITIES['primitive_skills'].keys()}
- Follow workflow:
    {ROBOT_CAPABILITIES['task_decomposition']['example_workflow']}

When receiving requests:
1. Parse request into semantic components
2. Cross-verify with spatial relationships
3. Generate executable skill sequence
```

```
4. Monitor environment changes for adaptation
"""
}
```

