# OpenReview forum: "DynTopo: Dynamic Topological Scene Graph for Robotic Autonomy in Human-Centric Environments"
_ICLR.cc/2026/Conference — ICLR 2026 Conference Withdrawn Submission_

### Official Review · Reviewer_4Vqq · 2025-10-30

**Soundness:** 2
**Presentation:** 3
**Contribution:** 2
**Rating:** 4
**Confidence:** 4

**Summary:**

The authors construct a Dynamic Topological Scene Graph (DynTopo) based on fixed-view cameras placed in various areas of an indoor scene. They leverage an LLM for task planning based on DynTopo to address long-term mobile manipulation tasks for robots in indoor environments, demonstrating certain robustness within a specific office scenario.

**Strengths:**

1.The method utilizes RGB-D images from both the robot's egocentric camera and fixed-view cameras to build a global static map for larger objects, while employing video frames from specific local regions to construct local dynamic maps for smaller, more frequently moved objects. This combination facilitates the robot's access to highly realistic and up-to-date scene information.

2.The approach encodes scene graph information into prompts suitable for LLMs, leveraging the LLM's capabilities to generate reasonable decision plans that guide the robot in accomplishing long-term mobile manipulation tasks.

**Weaknesses:**

1.The experimentation in this work is limited to having the robot execute a single task (delivering coffee). The lack of experiments with other tasks fails to adequately demonstrate the robustness of the DynTopo+LLM framework in diverse indoor scenarios.

2.The evaluation is conducted within only a single scene. This restricts the ability to claim strong generalization capabilities for the DynTopo+LLM approach across different environments.

**Questions:**

Experiments

1. This work integrates many existing algorithms with the goal of achieving general-purpose robotic autonomy, but the only experiment that truly supports this goal is a coffee delivery task in a single scene.

2. I believe the impact of the LLM on the overall task success rate is overlooked—for example, the influence of prompt design and the varying reasoning capabilities of different LLMs.

3. In Table2, the performance gap between DynTopo and static baselines appears minimal. A more in-depth analysis of this observation is warranted.

4. Embodied AI tasks should demonstrate efficiency. The authors do not report on computational or time efficiency metrics (e.g., SPL, inference speed), which are important for assessing practical applicability.

DYNTOPO

1. A major concern is the heavy reliance on dense deployment of environmental cameras to achieve comprehensive coverage of the entire space—essential for enabling robust autonomous navigation and handling occlusions or blind spots. Although the authors mention the potential use of an embodied camera, they neither specify a concrete strategy (e.g., how frequently a local graph should be constructed, or how local graphs from different time steps and viewpoints are managed and integrated) nor provide any experimental validation of this approach.

2. The paper lacks clarity on how real-time information is updated and integrated between the local dynamic maps and the global scene graph, e.g., the frequency at which the local graph is integrated into the global graph, and whether these updates are retained.

Details

1. The equation in line 263-264 is hard to understand, what are j and J? In line 256, is this matrix multiplication or Hadamard product?

---

### Official Review · Reviewer_KL4s · 2025-10-31

**Soundness:** 2
**Presentation:** 2
**Contribution:** 1
**Rating:** 2
**Confidence:** 4

**Summary:**

This paper proposes DynTopo, a Dynamic Topological Scene Graph framework for robotic autonomy. It integrates global static layouts and local dynamic relations, enabling robots to maintain spatial-temporal awareness and reason about evolving scenes using an LLM-based agent.

**Strengths:**

1) The paper identifies an important limitation of existing scene graph approaches—the inability to model dynamic object relationships in real time—and addresses it through a unified framework that integrates global layouts with local dynamic updates.
2) The real-world robotic experiments is valuable, showing that the proposed framework is not only conceptually sound but also practically viable for embodied autonomy in dynamic human environments.

**Weaknesses:**

1)	Limited originality: The proposed framework largely combines existing methods without introducing clear innovations. For example, the Topological Layouts Construction module (Sec. 4.1) directly adopts Topo-Field [Hou et al., 2025a] with only a predefined filtering rule; the Dynamic Relation Graph Generation (Sec. 4.2), which should be the core contribution, simply reuses FC-CLIP [Yu et al., 2023] for segmentation and PVSG [Yang et al., 2023b] for relation prediction, adding only a fixed weight matrix as the relation pair prior. Likewise, the Graph Fusion (Sec. 4.3) and LLM-driven Reasoning (Sec. 4.4) rely on manually specified rules and the widely used LLM-as-planner method. As a result, the overall framework appears as a composition of existing components and rules rather than a fundamentally new method.
2)	Insufficient experimental analysis: While the paper includes some quantitative comparisons and a real-robot demonstration, the experiments do not convincingly demonstrate the advantages of the proposed dynamic topology. For instance, Table 2 only compares static accuracy metrics, without any analysis of computational efficiency, or latency during dynamic updates—which are essential to justify the proposed design. More ablation or timing studies are needed to quantify how DynTopo improves upon static baselines in dynamic environments.
3)	Inconsistent terminology and unclear presentation: Several important terms are inconsistently used, which harms readability. For instance, “Relation Pair Prior” appears in Figure 2, Table 1 and 3 but is never formally defined in the main text—it only implicitly refers to the weight matrix in Sec. 4.2.

**Questions:**

1) In line 227, the paper mentions a “specific volume threshold" to distinguish large static objects from dynamic ones, but the exact value or selection rationale is not provided. How was this threshold determined, and does it generalize across environments? Similarly, in lines 260–261, the weighting coefficients (0.7 for human–furniture pairs and 0.3 for others) are fixed manually without empirical justification. Would it be possible to learn these parameters adaptively rather than assigning them?
2) The authors replace the PVSG ResNet-50 backbone with a ConvNeXt-Large CLIP model, but in Table 1 the ablation labeled "(w/o CNN-CLIP)" instead uses a ViT-based backbone. This makes it unclear whether the improvement truly comes from the new backbone or from other architectural differences. Could the authors provide a fairer evaluation?
3) The paper reports "R/mR@x" in Table 1, but these metrics are never defined or explained in the text. Could the authors explicitly state what "R" and "mR" stand for and how they are computed?

---

### Official Review · Reviewer_3Nji · 2025-11-01

**Soundness:** 3
**Presentation:** 3
**Contribution:** 2
**Rating:** 4
**Confidence:** 4

**Summary:**

The paper extends the traditional static topological scene graph to incorporate dynamic information for moving objects. Through tracking objects over time, the scene graph can be updated continuously to reflect the latest state of the environment. By representing not only structural relationships but also temporal interactions and motion patterns, the proposed dynamic graph enables more accurate and real-time scene understanding. This dynamic representation can be beneficial for downstream robotic applications such as navigation, manipulation, and task planning, where reasoning about how object relationships evolve is important.

**Strengths:**

* The overall system is well-designed, incorporating both static structure and dynamic object motion into the scene graph, enabling more complete and realistic scene understanding.
* The experiments in robotic tasks effectively demonstrate the usefulness and practicality of the proposed dynamic scene graph construction in real interaction scenarios.
* By leveraging spatiotemporal topology inference, open-set relation prediction, and an LLM-powered agent for reasoning, the method provides more expressive affordance understanding than traditional static scene graphs.

**Weaknesses:**

* Limited handling of unobserved dynamics:
 It remains unclear whether the method assumes all object movements must be continuously observed by the camera. The paper lacks discussion on how the system handles occlusions, partially observed dynamics, or object interactions occurring outside the camera’s field of view—situations commonly encountered in practical robotic environments.

* Unclear complexity of interaction scenarios:
 The interaction settings in experiments are insufficiently detailed. It is not explicitly stated how many objects are manipulated, what types of actions are executed, and whether multiple agents (robots or humans) interact simultaneously. Without such information, the robustness and scalability of the system in complex multi-object, multi-agent environments remain uncertain.


* Incomplete discussion of related dynamic scene graph work in robotics:
The paper does not sufficiently discuss closely related research in dynamic scene graphs for robotic manipulation, such as RoboEXP: Action-Conditioned Scene Graph via Interactive Exploration for Robotic Manipulation and Dynamic Open-Vocabulary 3D Scene Graphs for Long-Term Language-Guided Mobile Manipulation. A deeper comparison of contributions, assumptions, and application domains would help clarify the unique advantages and limitations of this work.

**Questions:**

* Can the proposed system handle scenes involving multiple simultaneous interactions, such as multiple objects being manipulated at once?

* How does the method cope with object movements that occur outside the robot’s field of view or during extended occlusions? Is there any mechanism to infer or recover missing dynamic information when an object temporarily leaves the camera’s coverage?

---

### Official Review · Reviewer_5NDe · 2025-11-01

**Soundness:** 3
**Presentation:** 3
**Contribution:** 3
**Rating:** 4
**Confidence:** 4

**Summary:**

This paper proposes to leverage scene graphs to capture 3D environments for downstream mobile manipulation tasks. Specifically, it exploits both high-level semantics of objects that are rarely moved and structural constrains over the graphs. A large language model is used to generate robot primitive actions based on scene graphs.

**Strengths:**

1: Mobile manipulation in large-scale environments is an interesting yet challenging research question.

2: The paper introduces several baselines and ablation studies. The proposed method is verified on a real-world mobile manipulator.

**Weaknesses:**

1: The generation of relational scene graphs has been extensively studied in the literature. This paper overlooks several important related work [1-6]; I encourage the authors to discuss how their method differs from these prior approaches.

2: This paper assumes access to known camera poses, object segmentations, and accurate depth. How can these inputs be reliably obtained in a real mobile manipulation setup? Furthermore, how robust is the system to noise in camera poses, segmentation errors, and depth inaccuracy?

3: This paper relies on predefined relations and skills. Are these relations and skills manually specified? If so, please provide more details on the design.

4: The approach depends on LLMs for generating robot actions. To what extent can the LLM generalize across different environments with carrying numbers, shapes, and sizes of objects?

References:

[1]: C. Agia, K. M. Jatavallabhula, M. Khodeir, O. Miksik, V. Vineet, M. Mukadam, L. Paull, and F. Shkurti. Taskography: Evaluating robot task planning over large 3D scene graphs. In Conference on Robot Learning, pages 46–58. PMLR, 2022.

[2]: Y. Huang, C. Agia, J. Wu, T. Hermans, and J. Bohg, “Points2plans: From point clouds to long-horizon plans with composable relational dynamics,” in 2025 IEEE International Conference on Robotics and Automation (ICRA), 2025.

[3]: Y. Zhu, A. Lim, P. Stone, and Y. Zhu. Vision-based manipulation from single human video with open-world object graphs. arXiv preprint arXiv:2405.20321, 2024.

[4]: Y. Huang, N. C. Taylor, A. Conkey, W. Liu, and T. Hermans, “Latent Space Planning for Multi-Object Manipulation with Environment-Aware Relational Classifiers,” IEEE Transactions on Robotics (T-RO), 2024.

[5]: Y. Wang, L. Fermoselle, T. Kelestemur, J. Wang, and Y. Li, “CuriousBot: Interactive Mobile Exploration via Actionable 3D Relational Object Graph,” arXiv preprint arXiv:2501.13338, 2025

[6]: K. Mo et al., “Structurenet: Hierarchical graph networks for 3D shape generation,” ACM Trans. Graph., vol. 38, no. 6, pp. 242:1–242:19, 2019.

**Questions:**

See the weakness section.

---

### Note · Authors · 2026-01-26

I have read and agree with the venue's withdrawal policy on behalf of myself and my co-authors.

---

### Meta-Review · Area_Chair_htgW · 2025-12-19

**Summary:**

The authors did not provide a rebuttal in response to the reviewers’ feedback. So, the concerns identified in the reviews remain unaddressed. In light of these unresolved issues, the Area Chair agrees with the reviewers’ recommendations and recommends rejection.

**Reviewer Concerns:**

No rebuttal was submitted to address the concerns.

**Reviewer Scores:**

There is no rebuttal, so the scores would not change.

---

### Decision · Program_Chairs · 2026-01-26

Reject